# GENERATIVE TIME-SERIES MODELING WITH FOURIER FLOWS

**Ahmed M. Alaa**
University of California, Los Angeles, USA
ahmedmalaa@ucla.edu

**Alex J. Chan**
University of Cambridge, UK
ajc340@cam.ac.uk

**Mihaela van der Schaar**
University of Cambridge, UK
University of California, Los Angeles, USA
Cambridge Center for AI in Medicine, UK
The Alan Turing Institute, UK
mv472@cam.ac.uk

## ABSTRACT

Generating synthetic *time-series* data is crucial in various application domains, such as medical prognosis, wherein research is hamstrung by the lack of access to data due to concerns over privacy. Most of the recently proposed methods for generating synthetic time-series rely on *implicit likelihood* modeling using generative adversarial networks (GANs)—but such models can be difficult to train, and may jeopardize privacy by "memorizing" temporal patterns in training data. In this paper, we propose an *explicit likelihood* model based on a novel class of normalizing flows that view time-series data in the *frequency-domain* rather than the time-domain. The proposed flow, dubbed a *Fourier flow*, uses a discrete Fourier transform (DFT) to convert variable-length time-series with arbitrary sampling periods into fixed-length spectral representations, then applies a (data-dependent) *spectral filter* to the frequency-transformed time-series. We show that, by virtue of the DFT analytic properties, the Jacobian determinants and inverse mapping for the Fourier flow can be computed efficiently in linearithmic time, without imposing explicit structural constraints as in existing flows such as NICE (Dinh et al. (2014)), RealNVP (Dinh et al. (2016)) and GLOW (Kingma & Dhariwal (2018)). Experiments show that Fourier flows perform competitively compared to state-of-the-art baselines.

## 1 INTRODUCTION

Lack of access to data is a key hindrance to the development of machine learning solutions in application domains where data sharing may lead to privacy breaches (Walonoski et al. (2018)). Areas where this problem is most conspicuous include *medicine*, where access to (highly-sensitive) clinical data is stringently regulated by medical institutions; such strict regulations undermine scientific progress by hindering model development and reproducibility. Generative models that produce sensible and realistic synthetic data present a viable solution to this problem—artificially-generated data sets produced by such models can be shared widely without privacy concerns (Buczak et al. (2010)).

In this paper, we focus on the *time-series* data setup, where observations are collected sequentially over arbitrary periods of time with different observation frequencies across different features. This general data setup is pervasive in the medical domain—it captures the kind of data maintained in electronic health records (Shickel et al. (2017)) or collected in intensive care units (Johnson et al. (2016)). While many machine learning-based predictive models that capitalize on such data have been proposed over the past few years (Jagannatha & Yu (2016); Choi et al. (2017); Alaa & van der Schaar (2019)), much less work has been done on generative models that could emulate and synthesize these data sets.

Existing generative models for (medical) time-series are based predominantly on *implicit likelihood* modeling using generative adversarial networks (GANs), e.g., Recurrent Conditional GAN (RCGAN) (Esteban et al. (2017)) and TimeGAN (Yoon et al. (2019)). These models apply representation learning

via recurrent neural networks (RNN) combined with adversarial training in order to map noise sequences in a latent space to synthetic sequential data in the output space. Albeit capable of flexibly learning complex representations, GAN-based models can be difficult to train (Srivastava et al. (2017)), especially in the complex time-series data setup. Moreover, because they hinge on implicit likelihood modeling, GAN-based models can be hard to evaluate quantitatively due to the absence of an explicitly computable likelihood function. Finally, GANs are vulnerable to training data memorization (Nagarajan et al. (2018))—a problem that would be exacerbated in the temporal setting where memorizing only a partial segment of a medical time-series may suffice to reveal a patient's identify, which defeats the original purpose of using synthetic data in the first place.

Here, we propose an alternative *explicit likelihood* approach for generating time-series data based on a novel class of normalizing flows which we call *Fourier flows*. Our proposed flow-based model operates on time-series data in the *frequency-domain* rather than the *time-domain*—it converts variable-length time-series with varying sampling rates across different features to a fixed-size spectral representation using the discrete Fourier transform (DFT), and then learns the distribution of the data in the frequency domain by applying a chain of data-dependent *spectral filters* to frequency-transformed time-series.

Using the *convolution property* of DFT (Oppenheim (1999)), we show that spectral filtering of a time-series in the frequency-domain—an operation that mathematically resembles affine transformations used in existing flows (Dinh et al. (2016))—is equivalent to a convolutional transformation in the time-domain. This enhancement in the richness of distributions learned by our flow comes at no extra computational cost: using Fast Fourier Transform (FFT) algorithms, we show that the entire steps of our flow run in $\mathcal{O}(T \log T)$ time, compared to the polynomial complexity of $\mathcal{O}(T^2)$ for a direct, time-domain convolutional transformation. We also show that, because the DFT is a linear transform with a *Vandermonde* transformation matrix, computation of its Jacobian determinant is trivial. The zero-padding and interpolation properties of DFT enables a natural handling of variable-length and inconsistently-sampled time-series. Unlike existing explicit-likelihood models for time-series data, such as deep state-space models (Krishnan et al., 2017; Alaa & van der Schaar, 2019), our model can be optimized and assessed through the *exact* likelihood rather than a variational lower bound.

## 2 PROBLEM SETUP

We consider a general temporal data setup where each instance of a (discrete) time-series comprises a sequence of vectors $\boldsymbol{x} = [\boldsymbol{x}_0, \dots, \boldsymbol{x}_{T-1}]$, $\boldsymbol{x}_t \in \mathcal{X}$, $\forall \, 0 \leq t \leq T-1$, covering a period of $T$ time steps. We assume that each dimension in the feature vector $\boldsymbol{x}_t$ is sampled with a different rate, i.e., at each time step $t$, the observed feature vector is $\boldsymbol{x}_t = [\, x_{t,1}[r_1], \dots, x_{t,D}[r_D] \,]$, where $r_d \in \mathbb{N}_+$ is the sampling period of feature dimension $d \in \{1, \dots, D\}$. That is, for a given sampling period $r_d$, we observe a value of $x_{t,d}$ every $r_d$ time steps, and observe a missing value (denoted as *) otherwise, i.e.,

$$x_{t,d}[r_d] \triangleq \begin{cases} x_{t,d}, & t \bmod r_d = 0, \\ *, & t \bmod r_d \neq 0. \end{cases} \tag{1}$$

The data setup described above is primarily motivated by medical time-series modeling problems, wherein a patient's clinical measurements and bio-markers are collected over time at different rates (Johnson et al. (2016); Jagannatha & Yu (2016)). Despite our focus on medical data, our proposed generative modeling approach applies more generally to other applications, such as speech synthesis (Prenger et al. (2019)) and financial data generation (Wiese et al. (2020)).

Each realization of the time-series $\boldsymbol{x}$ is drawn from a probability distribution $\boldsymbol{x} \sim p(\boldsymbol{x})$. In order to capture variable-length time-series (common in medical problems), the length $T$ of each sequence is also assumed to be a random variable—for notational convenience, we absorb the distribution of $T$ into $p$. One possible way to represent the joint distribution $p(\boldsymbol{x})$ is through the factorization:[1]

$$p(\boldsymbol{x}) = p(\boldsymbol{x}_0, \dots, \boldsymbol{x}_{T-1}, T) = p(T) \cdot \prod_{t=0}^{T-1} p(\boldsymbol{x}_t \mid \boldsymbol{x}_0, \dots, \boldsymbol{x}_{t-1}, T). \tag{2}$$

We assume that the sampling period $r_d$ for each feature $d$ is fixed for all realizations of $\boldsymbol{x}$. The feature space $\mathcal{X}$ is assumed to accommodate a mix of continuous and discrete variables on its $D$ dimensions.

---

[1]Our proposed method is not restricted to any specific factorization of $p(\boldsymbol{x})$.

**Key objective.** Using a training data set $\mathcal{D} = \{\boldsymbol{x}^{(i)}\}_{i=1}^n$ comprising $n$ time-series, our goal is to (1) estimate a density function $\hat{p}(\boldsymbol{x})$ that best approximates $p(\boldsymbol{x})$, and (2) sample synthetic realizations of the time-series $\boldsymbol{x}$ from the estimated density $\hat{p}(\boldsymbol{x})$. When dealing with data sets with variable lengths for the time-series, we model the distribution $p(T)$ independently following the factorization in (2). We model $p(T)$ using a binomial distribution. Throughout the paper, we focus on developing a flow-based model for the conditional distribution $p(\boldsymbol{x}_0, \ldots, \boldsymbol{x}_{T-1} \,|\, T)$.

## 3 PRELIMINARIES

Let $\boldsymbol{z} \in \mathbb{R}^D$ be a random variable with a known and tractable probability density function $p(\boldsymbol{z})$, and let $g : \mathbb{R}^D \to \mathbb{R}^D$ be an invertible and differentiable mapping with an inverse mapping $f = g^{-1}$. Let $x = g(\boldsymbol{z})$ be a transformed random variable—the probability density function $p(\boldsymbol{x})$ can be obtained using the change of variable rule as $p(\boldsymbol{x}) = p(\boldsymbol{z}) \cdot |\det \boldsymbol{J}[g]|^{-1} = p(f(\boldsymbol{x})) \cdot |\det \boldsymbol{J}[f]|$, where $\boldsymbol{J}[f]$ and $\boldsymbol{J}[g]$ are the Jacobian matrices of functions $f$ and $g$, respectively (Durrett (2019)).

Normalizing flows are compositions of $M$ mappings that transform random draws from a predefined distribution $\boldsymbol{z} \sim p(\boldsymbol{z})$ to a desired distribution $p(\boldsymbol{x})$. Formally, a flow comprises a chain of bijective maps $g = g^{(1)} \circ g^{(2)} \circ \cdots \circ g^{(M)}$ with an inverse mapping $f = f^{(1)} \circ f^{(2)} \circ \cdots \circ f^{(M)}$. Using the change of variables formula described above, and applying the chain rule to the Jacobian of the composition, the log-likelihood of $\boldsymbol{x}$ can be written as (Rezende & Mohamed (2015)):

$$\log p(\boldsymbol{x}) = \log p(\boldsymbol{z}) + \sum_{m=1}^{M} \log |\det \boldsymbol{J}[f_m]|. \tag{3}$$

Existing approaches to generative modeling with normalizing flows construct composite mappings $g$ with structural assumptions that render the computation of the Jacobian determinant in (3) viable. Examples of such structurally-constrained mappings include: Sylvester transformations, with a Jacobian corresponding to a perturbed diagonal matrix (Rezende & Mohamed (2015)), $1 \times 1$ convolutions for cross channel mixing, which exhibit a block diagonal Jacobian (Kingma & Dhariwal (2018)), and affine coupling layers that correspond to triangular Jacobian matrices (Dinh et al. (2016)).

### 3.1 FOURIER TRANSFORM

The Fourier transform is a mathematical operation that converts a finite-length, regularly-sampled time domain signal $\boldsymbol{x}$ to its *frequency domain* representation $\boldsymbol{X}$ (Bracewell & Bracewell (1986)). A $T$-point discrete Fourier transform (DFT), denoted as $\boldsymbol{X} = \mathcal{F}_T\{\boldsymbol{x}\}$, transforms a (complex-valued) time-stamped sequence $\boldsymbol{x} \triangleq \{x_0, \ldots, x_{T-1}\}$ into a length-$T$ sequence of (complex-valued) *frequency components*, $\boldsymbol{X} \triangleq \{X_0, \ldots, X_{T-1}\}$, through the following operation (Oppenheim (1999)):

$$X_k = \sum_{t=0}^{T-1} x_t \cdot e^{-2\pi j \cdot \frac{kt}{T}}, \ \forall 1 \leq k \leq T-1, \tag{4}$$

where $j$ corresponds to the imaginary unit of a split-complex number. Using Euler's formula, the complex exponential terms in (4) can be expressed as $e^{-2\pi j \cdot \frac{kt}{T}} = \cos\left(2\pi \cdot \frac{kt}{T}\right) - j \cdot \sin\left(2\pi \cdot \frac{kt}{T}\right)$. Thus, the Fourier transform decomposes any time-series into a linear combination of sinusoidal signals of varying frequencies—the resulting sequence of frequency components, $\boldsymbol{X}$, corresponds to the coefficients assigned to the different frequencies of the sinusoidal signals constituting the time domain signal $\boldsymbol{x}$. The DFT is a key computational and conceptual tool in many practical applications involving digital signal processing and communications (Oppenheim (1999)).

### 3.2 FOURIER TRANSFORM PROPERTIES

We will rely in developing our model on various key properties of the DFT. These properties describe various operations on the time-domain data and their dual (equivalent) operations in the frequency domain. The DFT properties relevant to the development of our model are listed as follows:

**Convolution:** $\boldsymbol{x}_1 \otimes \boldsymbol{x}_2 \iff \boldsymbol{X}_1 \odot \boldsymbol{X}_2.$

**Symmetry:** If $\boldsymbol{x}$ is real-valued $\Rightarrow X_k = X_{k-m}^*, \forall m \in \mathbb{N}.$

**Even/Odd Transforms:** $\mathcal{F}\{\text{Even}(\boldsymbol{x})\} = \text{Re}(\boldsymbol{X}), \ \mathcal{F}\{\text{Odd}(\boldsymbol{x})\} = \text{Im}(\boldsymbol{X}),$

where $\odot$ denotes element-wise multiplication, $\otimes$ denotes circular convolution, $\mathrm{Re}(.)$ and $\mathrm{Im}(.)$ denote the real and imaginary components, $\mathrm{Even}(\boldsymbol{x}) = (\boldsymbol{x} + \boldsymbol{x}_-)/2$, $\mathrm{Odd}(\boldsymbol{x}) = (\boldsymbol{x} - \boldsymbol{x}_-)/2$, where $\boldsymbol{x}_-$ signifies the reflection of $\boldsymbol{x}$ with respect to the $\boldsymbol{x} = 0$ axis, and $\boldsymbol{x} = \mathrm{Even}(\boldsymbol{x}) + \mathrm{Odd}(\boldsymbol{x})$. Another property that is relevant to our model is the *interpolation* property, which posits that zero-padding of $\boldsymbol{x}$ in the time domain corresponds to an up-sampled version of $\boldsymbol{X}$ in the frequency domain.

Table 1: The two main layers in an $N$-point Fourier flow, their inverses, and the corresponding log-determinant-Jacobian. The input to the flow, $\boldsymbol{x}$, is a $D \times T$ matrix comprising a set of $D$ time-series each of length $T$, and the output, $\boldsymbol{Y}$, is a $2 \times D \times N/2$ tensor comprising the filtered spectral representation of $\boldsymbol{x}$. Here, we show the application of the Fourier transform layer to a given feature dimension $d$—the same operation is applied independently to all feature dimensions. The vector $\widehat{\boldsymbol{X}}^{-d}$ is the reversed conjugate of $\widehat{\boldsymbol{X}}^d$, and $\boldsymbol{H} = [H_{i,j}]_{i,j}$. When cascading multiple Fourier flows, we alternate between feeding either of the real and imaginary channels of $\widehat{\boldsymbol{X}}$, $\mathrm{Im}(\widehat{\boldsymbol{X}})$ and $\mathrm{Re}(\widehat{\boldsymbol{X}})$, to the BiRNN network in the different flows within the cascade.

| Layer | Function | Inverse function | $\log\|\det \boldsymbol{J}\|$ |
|---|---|---|---|
| *Fourier transform* | $\bar{\boldsymbol{x}}^d = \boldsymbol{x}^d \cup \boldsymbol{0}_{N-T},$ $\bar{x}_{t,d} \leftarrow 0, \forall\, t \bmod r_d \neq 0,$ $\bar{\boldsymbol{X}}^d = \mathcal{F}_N\{\bar{\boldsymbol{x}}^d\},$ $\widehat{\boldsymbol{X}}^d = [\,\bar{X}_{0,d}, \ldots, \bar{X}_{0,\lceil N/2\rceil}\,].$ | $\bar{\boldsymbol{X}}^d = [\,\widehat{\boldsymbol{X}}^d, \widehat{\boldsymbol{X}}^{-d}\,],$ $\bar{\boldsymbol{x}}^d = \mathcal{F}_N^{-1}\{\bar{\boldsymbol{X}}^d\},$ $\bar{x}_{t,d} \leftarrow *, \forall\, t \bmod r_d \neq 0,$ $\boldsymbol{x}^d = [\,\bar{x}_0^d, \ldots, \bar{x}_{T-1}^d\,].$ | $\log\|\det W\| = 0.$ |
| *Spectral filtering* | $(\log \boldsymbol{H}, \boldsymbol{\mu}) = \mathtt{BiRNN}(\mathrm{Im}(\widehat{\boldsymbol{X}})),$ $\boldsymbol{Y}_1 = \boldsymbol{H} \odot \mathrm{Re}(\widehat{\boldsymbol{X}}) + \boldsymbol{M},$ $\boldsymbol{Y}_2 = \mathrm{Im}(\widehat{\boldsymbol{X}}),$ $\boldsymbol{Y} = \mathtt{concat}(\boldsymbol{Y}_1, \boldsymbol{Y}_2).$ | $\boldsymbol{Y}_1, \boldsymbol{Y}_2 = \mathtt{split}(\boldsymbol{Y}),$ $(\log \boldsymbol{H}, \boldsymbol{\mu}) = \mathtt{BiRNN}(\boldsymbol{Y}_2),$ $\mathrm{Re}(\widehat{\boldsymbol{X}}) = (\boldsymbol{Y}_1 - \boldsymbol{M})/\boldsymbol{H},$ $\widehat{\boldsymbol{X}} = \mathrm{Re}(\widehat{\boldsymbol{X}}) + j\,\mathrm{Im}(\widehat{\boldsymbol{X}}).$ | $\sum_{i,j} \log(\|H_{i,j}\|).$ |

## 4 FOURIER FLOWS

We propose a new flow for time-series data, the *Fourier Flow*, which hinges on the frequency domain view of time-series data (explained in Section 3.1), and capitalizes on the Fourier transform properties discussed in Section 3.2. In a nutshell, a Fourier Flow comprises two steps: (a) a *frequency transform* layer (Section 4.1), followed by (b) a data-dependent *spectral filtering* layer (Section 4.2). The steps involved in a Fourier Flow are summarized in Table 1 and Figure 1.

### 4.1 FOURIER TRANSFORM LAYER

In the first step of the proposed flow, we transform the time-series $\boldsymbol{x} = [\,\boldsymbol{x}_0, \ldots, \boldsymbol{x}_{T-1}\,]$ into its spectral representation via Fourier transform—we do so by applying the DFT operation (described in Section 3.1) to each feature dimension $d$ independently. Let $\boldsymbol{x}^d = [\,x_{0,d}[r_d], \ldots, x_{T-1,d}[r_d]\,]$ be the time-series associated with feature dimension $d$; the Fourier transform layer computes the $N$-point DFT of $\boldsymbol{x}^d$ for all $d \in \{1, \ldots, D\}$ through the following three steps:

$$\begin{aligned}
\textbf{Temporal zero padding:} \quad & \bar{\boldsymbol{x}}^d = \boldsymbol{x}^d \cup \boldsymbol{0}_{N-T},\ \bar{x}_{t,d}[r_d] = 0, \forall t,\ t \bmod r_d \neq 0, \\
\textbf{Fourier Transform:} \quad & \bar{\boldsymbol{X}}^d = \mathcal{F}_N\{\bar{\boldsymbol{x}}^d\}, \\
\textbf{Spectral cropping:} \quad & \widehat{\boldsymbol{X}}^d = [\,\bar{X}_{0,d}, \ldots, \bar{X}_{0,\lceil N/2\rceil}\,].
\end{aligned} \tag{5}$$

Here, $\boldsymbol{0}_{N-T}$ denotes a set of $N-T$ zeros, and the union $\cup$ denotes the padding operation that appends the zeros $\boldsymbol{0}_{N-T}$ to the time-series $\boldsymbol{x}^d$. The temporal zero-padding step capitalizes on the frequency interpolation and sampling properties of DFT (Section 3.2) to ensure that the padded time-series $\bar{\boldsymbol{x}}^d$ and its frequency spectrum $\bar{\boldsymbol{X}}^d$ have a fixed (predetermined) length of $N$, irrespective of the interval length $T$ and the sampling period $r_d$. Because the DFT coefficients are complex-valued, $\bar{\boldsymbol{X}}^d$ is a tensor with dimensions $2 \times 1 \times N$, and the collection of Fourier transforms for the $D$ feature dimensions, $\bar{\boldsymbol{X}}$, is a tensor with dimensions $2 \times D \times N$. That is, the DFT layer converts each time-series $\boldsymbol{x}$ into a two-channel, "image-like" $D \times N$ matrices $\mathrm{Re}(\bar{\boldsymbol{X}})$ and $\mathrm{Im}(\bar{\boldsymbol{X}})$ as shown in Figure 1. A flow with an $N$-point DFT will be referred to as an $N$-point Fourier flow in the rest of the paper. To guarantee a lossless recovery of the time-series $\boldsymbol{x}$ via inverse DFT, we ensure that $N \geq T$.

Finally, the *spectral cropping* step in (5) discards the $N/2+1^{th}$ to $N^{th}$ frequency components in both $\text{Re}(\bar{\boldsymbol{X}})$ and $\text{Im}(\bar{\boldsymbol{X}})$. This is because $\boldsymbol{x}$ is real-valued, hence $\text{Re}(\bar{\boldsymbol{X}})$ and $\text{Im}(\bar{\boldsymbol{X}})$ are symmetric and anti-symmetric, respectively (See Section 3.2), which renders the discarded frequency components redundant. The final output of this layer, $\widehat{\boldsymbol{X}}$, is a $2 \times D \times N/2$ tensor.

**DFT computation.** Since the DFT operation in (4) is linear, we can represent the frequency transform step in (5) through a linear transformation and apply DFT via matrix multiplication as follows:

$$\bar{\boldsymbol{X}}^d = W\bar{\boldsymbol{x}}^d, \;\; W = \frac{1}{\sqrt{N}} \begin{bmatrix} 1 & 1 & 1 & \cdots & 1 \\ 1 & \omega & \omega^2 & \cdots & \omega^{N-1} \\ 1 & \omega^2 & \omega^4 & \cdots & \omega^{2(N-1)} \\ \vdots & \vdots & \vdots & \ddots & \vdots \\ 1 & \omega^{N-1} & \omega^{2(N-1)} & \cdots & \omega^{(N-1)(N-1)} \end{bmatrix}, \text{ and } \omega = e^{-2\pi j/N}. \quad (6)$$

When $N$ is set to be a power of 2, the DFT operation in (6) can be implemented using any variant of the Fast Fourier Transform (FFT) methods (Nussbaumer (1981); Van Loan (1992)), such as the Cooley-Tukey FFT algorithm (Cooley & Tukey (1965)), with a computational complexity of $O(N \log N)$.

**Determinant of the DFT Jacobian.** The DFT is a natural and intuitive transformation for temporal data, but how does introducing the DFT mapping affect the complexity of the Jacobian determinant of the flow? To answer this question, we note that the DFT matrix $W$ in (6) is a (square) *Vandermonde* matrix, thus we can evaluate the DFT Jacobian determinant in closed-form as follows:

$$| \det(\boldsymbol{J}[W]) | \stackrel{(a)}{=} | \det(W) | \stackrel{(b)}{=} \left| \left( \frac{1}{\sqrt{N}} \right)^N \prod_{1 < n < m \leq N} (\omega^m - \omega^n) \right| \stackrel{(c)}{=} 1, \quad (7)$$

where (a) follows from the fact that the Jacobian of a linear transformation is equal to the transformation matrix $W$, and (b) follows from the scalar multiplication property of determinants, which posits that $\det(\alpha V_{N \times N}) = \alpha^N \det(V_{N \times N})$ for a scalar $\alpha$ and matrix $V$, combined with the formula for the determinants of Vandermonde matrices (Björck & Pereyra (1970)). (c) follows from the fact that the magnitude of the complex exponential is $|\omega| = 1$, and the polynomial product in (7) comprises a total of $N^{N/2}$ terms, each of which is of the form $|\omega^n \omega^m| = 1$. The result in (7) shows that the DFT mapping does not incur extra computational costs when evaluating the flow likelihood in (3).

## 4.2 SPECTRAL FILTERING LAYER

The second layer of the Fourier flow is an *affine coupling* layer, similar to that originally introduced in (Dinh et al. (2014; 2016)), but applied to the time-series $\boldsymbol{x}$ in the frequency domain as follows:

$$(\log \boldsymbol{H}, \boldsymbol{M}) = \texttt{BiRNN}(\text{Im}(\widehat{\boldsymbol{X}})), \; \boldsymbol{Y}_1 = \boldsymbol{H} \odot \text{Re}(\widehat{\boldsymbol{X}}) + \boldsymbol{M}, \; \boldsymbol{Y}_2 = \text{Im}(\widehat{\boldsymbol{X}}), \; \boldsymbol{Y} = \texttt{concat}(\boldsymbol{Y}_1, \boldsymbol{Y}_2),$$

where $\boldsymbol{H}$ and $\boldsymbol{\mu}$ are $D \times N$ matrices, $\texttt{BiRNN}$ denotes a bi-directional recurrent neural network (Schuster & Paliwal (1997)), and $\odot$ denotes the Hadamard (element-wise) product. Here, we split the real and imaginary channels in $\widehat{\boldsymbol{X}}$ rather than splitting the feature dimensions as in (Dinh et al. (2016)). The detailed steps of the flow and its inversion are given in Table 1.

The affine transformation in the frequency domain can be thought of as a spectral filtering operation whereby the frequency transform of the *even part* of the time-series, $\text{Re}(\widehat{\text{X}})$, is applied to a filter with a transfer function $\boldsymbol{H}$ (Recall the even-odd decomposition properties of DFT in Section 3.2). The transfer function itself is data-dependent: it depends on $\text{Im}(\widehat{\text{X}})$, or equivalently, the frequency transform of the *odd part* of the time-series $\boldsymbol{x}$. The mapping from $\text{Im}(\widehat{\text{X}})$ to the transfer function $\boldsymbol{H}$ is implemented through an RNN that shares parameters across all different frequency components, since $N$ can grow very large for time-series with a large $T$. We use a bi-directional RNN since all frequency components are available at our disposal at the *same time*.

## 4.3 FOURIER FLOWS IN THE TIME-DOMAIN

How does the Fourier flow mapping look like in the time domain? Using the convolution property of the DFT (Section 3.2), the time-domain view of the Fourier flow can be expressed as follows

$$\boldsymbol{y}_1 = \boldsymbol{h}(\bar{\boldsymbol{x}}) \otimes \text{Even}(\bar{\boldsymbol{x}}) + \boldsymbol{\mu}, \; \boldsymbol{y}_2 = \text{Odd}(\bar{\boldsymbol{x}}), \; \boldsymbol{y} = \texttt{concat}(\boldsymbol{y}_1, \boldsymbol{y}_2), \quad (8)$$

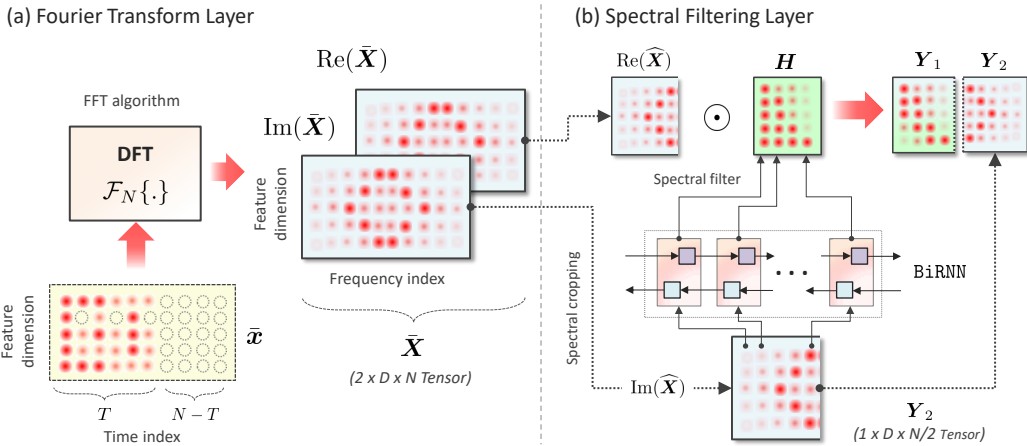

Figure 1: **Pictorial depiction of the two layers underlying a Fourier flow.** Here, we consider an exemplary instance of a time-series with $D = 5$ and $T = 6$, to which we apply a 10-point Fourier flow ($N = 10$). In all displays, the $x$-axis corresponds to either the time or frequency indexes, the $y$-axis corresponds to the feature dimension, and the red dots correspond to the value associated with a given time (frequency) index in a given feature dimension. (Darker shades of red correspond to higher values.) **(a) DFT layer.** In this example, we assume the sampling periods to be $r_1 = 2$, and $r_d = 1$, $\forall d \neq 1$. In the first step of the flow, we set all missing values in the under-sampled feature dimension to 0, and padd $N - T = 4$ zeros to all features. (Padded zeros are depicted as white dotted circles), which results in a padded time-series $\bar{x}$ in the form of a $D \times N$ matrix. Next, the DFT operation is applied to each row in $\bar{x}$, resulting in a $2 \times D \times N$ tensor $\bar{X}$ comprising the real and imaginary components of the frequency transform. **(b) Spectral filtering layer.** The imaginary component of the (cropped) spectrum is passed to a bi-directional RNN, which views the input as an $N/2$-length sequence of $D$-dimensional features. The RNN outputs a $N/2 \times D$ sequence that corresponds to the spectral filter $H$, which is then multiplied by $\mathrm{Re}(\widehat{X})$. The final output concatenates the filtered spectrum, $Y_1$, with $Y_2 = \mathrm{Im}(\widehat{X})$.

where $y_i = \mathcal{F}_N^{-1}\{Y_i\}$, $i \in \{1, 2\}$, $\mu = \mathcal{F}_N^{-1}\{M\}$, and $h = \mathcal{F}_N^{-1}\{H\}$ is the impulse response of the spectral filter $H$. That is, the Fourier flow in the time domain corresponds to a circular convolution between the even part of $\bar{x}$ and the inverse DFT of the filter $H$, which depends on the odd part of $\bar{x}$.

The time-domain view of a Fourier flow elucidates its *multifaceted* modeling advantages. First, there is a *representational* advantage of having a more expressive convolutional transformation in (8) compared to the element-wise affine transformations in the coupling layers of existing methods (Dinh et al. (2014; 2016); Kingma & Dhariwal (2018)). Moreover, a Fourier Flow does not require splitting feature dimensions disjointly, but rather decomposes each feature to its even and odd components. Second, this richer representation comes with no extra computational cost: while time-domain convolution would run in $\mathcal{O}(N^2 D^2)$ time (Hunt (1971)), our spectral filter applies an $\mathcal{O}(N)$ element-wise affine operation in the frequency domain, and the DFT runs in $\mathcal{O}(DN \log N)$ time. Computation of the Jacobian determinant is achieved by simply adding all elements of $H$ (Table 1) as it is the case in time-domain affine coupling methods. Finally, the DFT operation enables handling of variable-length time-series and inconsistent sampling periods across features without any extra modeling efforts.

**Related work.** Existing explicit-likelihood models for sequential data are based predominantly on state-space modeling approaches, which assume that hidden state dynamics control the sequence of observed data. Examples of such models include hidden Markov model (Beal et al. (2002)), deep Markov models (Krishnan et al. (2017)), and attentive state-space models (Alaa & van der Schaar (2019)). The key advantage of our model compared to these is that ours is optimized and assessed using the *exact* likelihood rather than a variational lower bound.

Existing models based on normalizing flows have been primarily focused on *static* (non-temporal) data (Ziegler & Rush (2019))—examples include NICE (Dinh et al., 2014), RealNVP (Dinh et al., 2016), and GLOW (Kingma & Dhariwal, 2018; Prenger et al., 2019). These flows assume highly-structured transformation with easy-to-compute Jacobian determinants, e.g., diagonal, block diagonal and triangular Jacobian matrices. Fourier flows assume a structured transformation in the *frequency domain*, which corresponds to a richer and more complex representation than existing flows in time domain. In particular, a Fourier flow can be thought of as the frequency-domain dual of the affine transformation

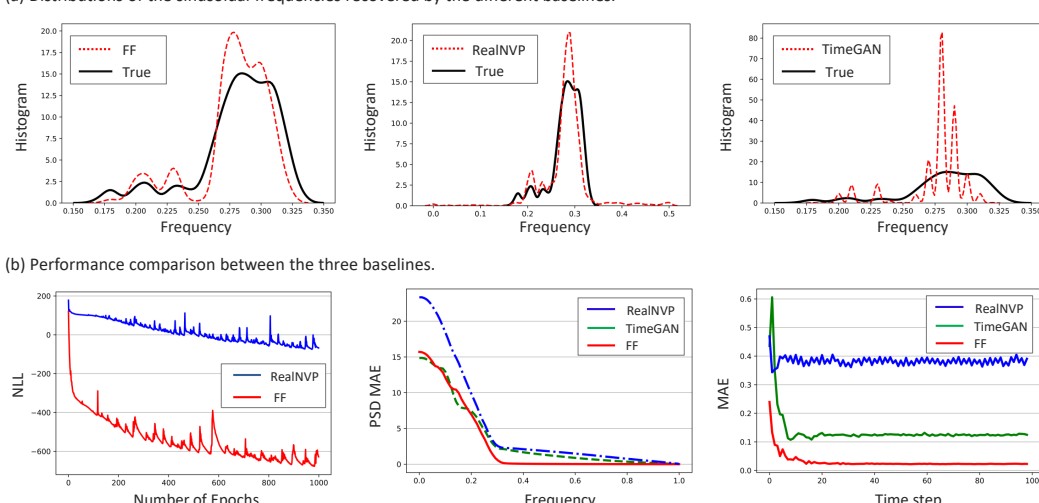

Figure 2: Performance of FF, RealNVP and TimeGAN in learning to generate random sinusoidal time-series with random frequencies. (a) Here, we estimate the dominant frequency of the sinusoidal signals generated by each baseline and plot the histogram for the estimated frequencies across that of the ground-truth distribution of the true frequency $f$. We plot a kernel density estimate of the histogram in each plot. Note that the ranges of the $x$- and $y$-axes in each plot change according to the estimated distributions. (b) Assessment of the time-series data generated by the different baselines in terms of exact data likelihood, spectral properties and predictive utility.

of RealNVP, which translates to a more complex convolutional mapping in the time domain. The closest work to ours is the *invertible convolutional flows* developed independently in (Karami et al. (2019)). This work does not deal with time-series data, but introduces a convolutional transformation on the feature dimension using Toeplitz matrices, and implements the transformation using DFT.

## 5 EXPERIMENTS

### 5.1 ILLUSTRATIVE RESULTS ON SYNTHETIC DATA

To examine the utility of our explicit-likelihood frequency-domain model, we start off by comparing Fourier flows with their time-domain analogues, RealNVP flows (Dinh et al. (2016)), and a state-of-the-art implicit-likelihood method, TimeGAN (Yoon et al. (2019)), in a stylized experimental setup. In particular, we consider the following time-series data generation process:

$$\boldsymbol{x} = \sin(ft + \phi), \ \phi \sim \mathcal{N}(0, 1), \ f \sim \text{Beta}(\alpha = 2, \beta = 2), \ t \in \{0, \ldots, T - 1\}, \tag{9}$$

where Beta denotes a Beta distribution with shape parameters $\alpha$ and $\beta$, and $T = 100$. That is, our time-series data is a set of sinusoidal sequences with random frequency and phase, where the phase is drawn from a normal distribution and the frequency is drawn from a Beta distribution. We generate a total of 1,000 time-series based on (9) and use these to train all baselines. We compare a Fourier flow (FF) model with a composition of 10 flows and a (single-layer) BiRNN with 200 hidden units with an equivalent RealNVP model with the same number of flows and an BiRNN for generating the coefficients of the affine transformation. For the TimeGAN model, we use the hyper-parameter configuration recommended in (Yoon et al. (2019)). We train all models with 1,000 iterations and a batch size of 128—we then generate 1,000 synthetic time-series from each trained model.

In Figure 2(a), we plot the histogram for the frequencies of the time-series generated by each model and compare it with the ground-truth Beta distribution of $f$. We estimate the frequency of each generated time-series (which are generally not perfectly sinusoidal) by finding the dominant frequency component $\hat{f}$ in the DFT of each generated time-series. As we can see, the distribution of $\hat{f}$ recovered by FF bears the strongest resemblance to the true Beta distribution. Moreover, we note that the implicit likelihood approach in TimeGAN concentrates the probability mass on few frequencies, indicating that it memorized some of the sinusoidals in training data, which is problematic in applications where

we would want to keep the training samples *private*—this problem does not arise in FF where the learned distribution is smooth and does not concentrate around any specific frequencies. On the other hand, RealNVP learned a distribution with a much wider support than that of the original data.

The richness of the (frequency-domain) FF transformation compared to the (time-domain) RealNVP is evident in the lower negative log-likelihood achieved by our model (Figure 2(b), left). In Figure 2(b) (middle), we compare all baselines with respect to the *power spectral density* (PSD) of the 1,000 time-series sampled from each model; the PSD is a spectral representation of a stochastic process given by $PSD = \mathcal{F}\{\mathbb{E}[\,\boldsymbol{x} \cdot \boldsymbol{x}_-^*\,]\}$. Figure 2(b) plots the cumulative mean absolute error (MAE) between the PSD of the original data and that of the samples generated by each model within all frequency bands $[0, f], \forall f \in [0, 1]$. FF provides competitive accuracy in the recovered PSD compared to TimeGAN (MAE: 2.83 for FF, 3.44 for TimeGAN), and significantly outperforms RealNVP (MAE: 5.12). This implies that the distribution of frequencies generated by FF accurately resembles the original data.

Finally, we evaluated the accuracy of the sampled data in the time-domain by assessing their predictive usefulness as follows: we trained a vanilla RNN model using each of the three synthetically generated samples to sequentially predict the next value in each time-series in the original data. Figure 2(c) (right) shows the MAE of the three RNN models across all time step—as we can see, the RNN trained on FF-generated data consistently outperforms the baselines.

Table 2: Performance scores for all baselines.

| Method | | (*F*-score, MAE) | |
| --- | --- | --- | --- |
| | Stocks | Energy | Lung cancer |
| TimeGAN | (0.938, 0.173) | (0.479, 0.056) | (0.732, 3.248) |
| WGAN | (0.989, 0.010) | (0.985, 0.049) | (0.517, 4.824) |
| ARIMA | (0.648, 0.016) | (0.667, 0.037) | (0.659, 4.299) |
| WaveNet | (0.441, 0.123) | (0.013, 0.213) | — |
| RealNVP | (0.979, 0.020) | (0.965, 0.053) | (0.866, 5.762) |
| FF | (0.984, 0.009) | (0.949, 0.039) | (0.904, 3.207) |

## 5.2 PERFORMANCE EVALUATION ON REAL DATA

We replicate the experimental setup in (Yoon et al. (2019)) to test the performance of our model on multiple time-series data sets with respect to its predictive utility, i.e., the extent to which models trained on the synthetic samples can perform well on prediction tasks in the original sample. We measure performance with respect to this criterion via a *predictive score* that corresponds to the accuracy of an RNN trained on the synthetic data to predict next step in real data. In addition, we measure the quality of the synthetic data generated by each baseline using the precision and recall metrics (Sajjadi et al. (2018)) averaged over all time steps, and then we combine both scores into a single *F*-score. We conduct experiments on: Google stocks data, UCI *Energy* data set, and a longitudinal follow-up clinical data set for lung cancer patients. To highlight the richness of the learned FF distributions, we also visualize the t-SNE plots for the FF samples compared to their time-domain analogues generated by RealNVP.

We compare Fourier Flows with the following baselines: TimeGAN (Yoon et al. (2019)), Wasserstein GAN (WGAN) trained on time steps as distinct feature dimensions, a Autoregressive Integrated Moving-Average (ARIMA) model, WaveNet (Oord et al. (2016)) and a naïve RealNVP benchmark with time steps treated as feature dimensions. Performance results are provided in Table 2—the results indicate that FF consistently generates high-quality synthetic time-series that are useful for predictive model training (as quantified by the predictive score). FF outperforms all methods with respect to the predictive score on Stocks and Lung cancer data sets, and its learned distribution more closely resemble the real distribution compared to the time-domain method (Figure 3 shows the t-SNE plots on Google's stock data). It is also worth mentioning that FF performs competitively with respect to the domain- and task-agnostic precision and recall with respect to all data sets, which means that the FF-generated synthetic samples could reliably replace original ones in a wide variety of modeling tasks.

(a) t-SNE plot for Fourier flows.

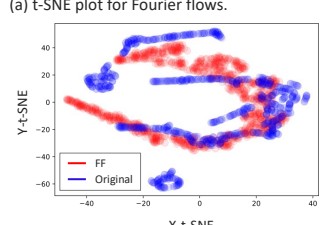

(b) t-SNE plot for RealNVP.

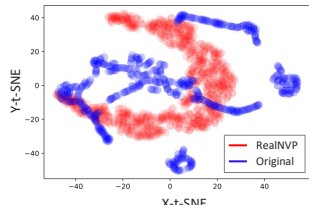

Figure 3: t-SNE plots for Stock.

## 6 DISCUSSION

In this paper, we introduced an explicit likelihood model based on a novel class of normalizing flows that models the distributions of stochastic time-series data in the frequency-domain rather than the time-domain. Capitalizing on the Fourier transform properties, the proposed flow naturally handles arbitrary sample rates and duration of time-series data, and enables learning richer representation compared with the existing methods at no extra computational cost.

Fourier flows enjoy *three* modeling advantages that enable them to accurately model time-series distributions. First, the DFT layer compresses temporal information into a low-dimensional spectral representation, enabling a more efficient distribution learning. Consider for instance the sinusoidal example in Section 5.1. In this example the data is drawn from a stochastic process $\sin(ft + \phi)$ where $f$ is a random frequency. To model length-$T$ time-series drawn from such process using conventional methods, we would need to model a $T$-dimensional random variable. However, using the DFT transform, we would be modeling the spectral representation $X = \frac{2}{j}[\delta(v + f) - \delta(v - f)]$ where $\delta$ is the Dirac-delta function. Thus, the spectral representation can be fully described with one piece of information, which is the location of the frequency component $f$, i.e. a 1-dimensional random variable. Second, Fourier flows enable construction of complex transformations at no extra cost for Jacobean computation. That is, a DFT layer followed by a simple transform (such as an affine transform) would amount for a complex overall transform without any extra complications associated with the Jacobean transformation. Finally, the usage of an RNN model in the novel spectral filtering layer captures the sequential nature of the spectral data.

Another key advantage for Fourier flows is that they are *explicit likelihood* models with a tractable exact likelihood. This means that they can be trained and evaluated using the exact model likelihood instead of a variational lower bound as in most existing approaches. Moreover, explicit likelihood models are hypothesized to be superior to implicit likelihood models, such as GANs, in terms of privacy preservation, as they are less likely to memorize training points. While the theoretical generalization properties of GANs have been previously investigated Nagarajan et al. (2018), less work has been done in studying the theoretical generalization performance of flows. Proving that normalizing flows do not memorize data is beyond the scope of this paper as it would amount for a broader and more general result that is not limited to the time series setup. Such theoretical analysis is an interesting subject for future work.

While Fourier transforms enjoy various computational advantages, they may fall short when modeling non-stationary time-series, where we know that the distribution of each new observation is not just a function of the process history, but also the time index. Thus, investigating generalizations of our model with more general spectral transforms, such as generalized Fourier transforms or discrete Wavelet transforms are promising directions for future work.

## ACKNOWLEDGMENTS

We would like to thank the reviewers for their valuable feedback. The research presented in this paper was supported by the US Office of Naval Research (ONR), and by the National Science Foundation (NSF) under grant numbers 1407712, 1462245, 1524417, 1533983, and 1722516.

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
