# OpenReview forum: "Generative Time-series Modeling with Fourier Flows"
_ICLR.cc/2021/Conference — ICLR 2021 Poster_

### Official Review · AnonReviewer1 · 2020-10-18
**A Time Series Flow-based Generative Model using DFT**

**Rating:** 5
**Confidence:** 4

**Review:**

### Summary

The paper presents Fourier Flows (FF), which is a time series generative model in the frequency domain. It shows that the Jacobian of the DFT is equal to 1, which means that DFT does not add too much overhead. The results on the real-world datasets are encouraging and expected because the predictive results mainly rely on the overall trend of the time series. By analysis in the frequency domain, we usually can capture the main trend accurately. The main concerns for the paper are the computational overhead on the proposed algorithm on non-periodic, long, and variable-length time series.



### Feedback

* The strongest point of the paper is when the authors show that the Jacobian of the DFT matrix is 1. Thus, taking DFT does not make the generative model much more complex.
* The other advantage of FF can be in handling missing values. Unfortunately, the authors do not expand on this aspect.
* The main issue with the FF is its computational complexity. The authors write: "To guarantee a lossless recovery of the time-series x via inverse DFT, we ensure that $N\geq T$." This means FF is impractical for long time series and quite inefficient in handling variable-length time series.
* Moreover, Fourier transforms are mainly intended for periodic signals and do not provide concise representations of non-stationary signals. This inefficiency is why Wavelets and DWT have been invented. The synthetic data generation model gives an unfair advantage to FF because it uses periodic signals.
* The authors' description for Eq. (8) is misleading. $\mathbf{H}$ and $\mathbf{h}$ are not fixed parameters, they are functions of the input. Thus, the model is a form of self-attention. I hope that the authors have done the training correctly.
*The authors criticize GANs for memorization but never show that FFs do not memorize data points.
* The choice of generating H using a BiRNN on only the $Im(X)$ is puzzling. Why didn't the authors use complex-valued RNNs to operate on both real and imaginary parts and not lose the phase information?
* On page 2, the authors discuss a parametric model for T, but they never elaborate further on it.
* "where $x_-$ signifies the reversed version of $x$": the authors should clarify that this is the reflection with respect to $x=0$ axis. Otherwise, we should add a time-shift operation too.

----
### Post-Response Update
I don't think the authors have answered my second set of questions. While there are some doubts remaining in the paper, the idea looks fine. Although, I think a new paper with DWT will outperform this approach soon. I do not change my vote at this time.

---

> ### Author Response · Authors · 2020-11-17
> **Response to comments by Reviewer #1**
>
> Thank you very much for your insightful comments and valuable suggestions. In what follows, we provide a point-by-point response to your comments, and explain the changes that will take place in the final version of the paper in order to reflect your suggestions.
>
> ***Clarifications***
>
> First off, we would like to address some of your concerns by clarifying some points that may have not been clear in the original version of this submission.
>
> 1) *Regarding equation (8) and the notation for $H$ and $h$*
>
> Thank you for pointing out to this typo. As you mention, $H$ and $h$ are not fixed parameters, but they should rather be denoted as $H(x)$ and $h(x)$ to avoid confusion. Of course, we have done the training using the spectral filtering architecture in Sec 4.2 and Fig 1.
>
> 2) *Do FFs memorize training data?*
>
> Please note that our criticism of generative models that can potentially memorize data points was directed at the **implicit density modeling approach** in general, and not just GANs. This is one of the key motivations for developing an **explicit density model** instead. Fourier flows, unlike GANs, are explicit density models that specify a class of probability distributions as its hypothesis class, and then selects the best distribution within this class to maximize a tractable and exact likelihoods. Because of its reliance on a maximum likelihood approach instead of a discriminator-generator architecture, normalizing flows has been shown to be less prone to data memorization [R1], and some previous works have even combined Flows and GANs to help "regularize" GANs' training and suppress its potential for data memorization [R2].
>
> While the theoretical generalization properties of GANs have been previously investigated (Nagarajan et al. (2018)), less work has been done in studying the theoretical generalization performance of flows. Proving that normalizing flows do not memorize data is beyond the scope of this paper as it would amount for a broader and more general result that does not just apply for Fourier flows or the time series setup. Please note though that we have shown (empirically) that FFs produce smoother distributions that do not concentrate probability mass on training points compared to TimeGANs (see Figure 2(a)).
>
> In the final version of this paper, we will make it clear that rigorous comparisons of data memorization potential in both classes of models is an important topic for future work, and we will be careful with phrasing the claims related to data memorization in the introductory section.
>
> 3) *Why BiRNN operates on $Im(X)$ only?*
>
> We can think of the complex number $Re(X) + j Im(X)$ (for a $D$-dimensional $X$) as a new feature with $2D$ dimensions. The spectral filtering layer in our model uses the same architecture of the affine coupling layer in RealNVP and GLOW: it uses half of the dimensions (which in this case is $Im(X)$) to generate the data-dependent filter $H$, and then applies this filter to the remaining dimensions (which in this case is $Re(X)$). The advantage of this scheme is that inversion is easy as the Jacobean of this transformation is a triangular matrix.
>
> **Please note that this transformation does not throw away the phase information.** In fact, the phase information is conveyed in $angle(X)$, which is split between both the real and imaginary values of $X$. Moreover, both $Re(X)$ and $Im(X)$ are present in the computation of $Y_1$ as highlighted in Table 1. Finally, and most importantly, when we cascade multiple flows, we alternate between generating the filter using $Re(X)$ and $Im(X)$ (and applying it to $Im(X)$ and $Re(X)$, respectively), so the spectral filter is not generated exculsively using  $Im(X)$ in all of the cascaded flows. This point was not mentioned in the original submission but we will highlight it in the final version of the paper.
>
> 4) *Other issues*
>
> - The sequence length $T$ is modeled through a simple binomial distribution. This was mentioned in page 2 but we will highlight it more prominently in the final version of the paper.
>
> - We will clarify that $x_{-}$ is a reflection with respect to the $x=0$ axis. Thank you for pointing out to this issue.
>
> - It is true that another advantage of FF is that it can be in handling missing values. Because the competing baselines do not have any inherent mechanisms for handling missing data, we did not consider missing data in our experiments to enable a fair comparison. However, we will add a discussion on the advantages of FFs when dealing with missing variables or irregulary sampled time series.

---

> > ### Author Response · Authors · 2020-11-18
> > **Response to comments by Reviewer #1 (Cont'd)**
> >
> > ***Computational complexity of Fourier Flows***
> >
> > Please note that *computational efficiency* is in fact one of the key advantages of Fourier flows, and one of the key motivations for using DFT (which can be efficiently implemented with FFT algorithms) in our flow transformation. To recapitulate the chief points pertaining to the computational efficiency of FFs, please recall the following:
> >
> > - With FFs, we can implement time domain convolution in linearithimic time $O(N log N)$, whereas implementing the same operation directly in the time domain without the DFT layer would be of an $O(N^2)$ complexity.
> >
> > - As we have shown in Sec 4, the DFT transform is unitary and hence its Jacobean matrix is trivial with an $O(1)$ complexity. Using another trainable transform that would provide the same representational complexity (achieved via time domain convolutions) as FF would cost a complexity of $O(N^3)$ for computing the Jacobean determinants.
> >
> > We understand that your concern pertains to the way we handle variable length: by ensuring that $N > T$ for lossless recovery, the cost of training would be unnecessarily high if the maximum length time series is much longer than the average length in the training data. However, we think that variable length time series can be very easily and efficiently handled in our model, and our flow construct is not in any way compromised by this issue for the following reasons:
> >
> > - First, note that the extra computational complexity of setting $N = T_{max}$ instead of $T_{avg}$ (where $T_{max}$ and $T_{avg}$ are the maximum and average lengths of the time series in the data) **only costs an extra linear complexity of $T_{max} - T_{avg}$**, which in most practical applications would be negligible.
> >
> > - More importantly, we note that because our spectral filtering layer uses and RNN, we do not have to fix the size of the FFT transform at all. **The flow can easily deal with variable length series simply by generating variable length $h$ and $H$ from the BiRNN layer for each training point.** This is in fact one key advantage of using an RNN instead of a feedfroward NN in the spectral filtering layer.
> >
> > - **The size of the FFT transform $N$ can also be varyed across different batches,** where in each batch we set $N$ to the maximum sequence length within each batch. This will minimze the extra computational cost associated with extremely variable lengths of sequences and a fixed FFT per training iteration.
> >
> > In the original submission, we adopted the naive solution of fixing $N > T$ because most data sets involved in our experiments do not have big variances in sequence lengths, and hence we opted to simplify the exposition of our model by making this simple modeling choice. However, this is not the only modeling choice one can make---**variable lengths sequences are not an impediment to our model as any of the three alternative modeling choices above can be used** to deal with large variability in sequence lengths more efficiently. **It is also important to note that this flexibility in modeling variable length sequences are only possible because of the usage of DFT (which is naturally capable of upsampling or downsampling time domain series), and the usage of BiRNN which is capable of generating variable length filters.**
> >
> > While we disagree that variable-length sequences hinder our model, we totally agree that the point you raised is important, hence we will clarify the alternative modeling choices stated above in the final version of the paper.

---

> > > ### Author Response · Authors · 2020-11-18
> > > **Response to comments by Reviewer #1 (Cont'd)**
> > >
> > >
> > > ***Non-stationary and non-periodic signals***
> > >
> > > Please note that the Fourier transform exists for any signal that is absolutely integrable. In the signal processing literature, these signals are known as energy signals. In fact, because all signals we deal with are of a finite-duration, none of these are periodic. The Fourier transform is not restricted to or intended for periodic signals only, neither does it fail to provide an invertible transform for any non-periodic signal **as long as it is integrable as all practical non-stationary signals are**.
> > >
> > > It is true, however, that our flow will be significantly more efficient in representing periodic signals such as the sinusoidal signal involved in our synthetic data experiments, or the ECG data involved in our extra experiments (Please refer to our response to Reviewer 4 for more details on this). **While a more concise representation of periodic and quasi-periodic signals is one of the advantages of our model, it is neither the only advantage nor the key one** (please refer to our response to Reviewer 4 where we list 3 advantages for using DFT). While this advantage manifested in the synthetic data experiments, please note that FFs performed competitively on real world data sets as well, where the data was non-periodic and non-stationary.
> > >
> > > Finally, extending our work using other spectral transforms such as Wavelets and DWT as you suggest is an excellent idea. In fact, our approach is general enough to accommodate any unitary spectral transform that enjoys the convolution-muliplication duality. A generalized Fourier transform learnable with learnable frequency scales and phase shift is also another option that can capture non-stationarity. Importantly, all of these alternatives will hinge on the same concepts and architectures presented in our paper, hence we view them as extensions rather than alternatives. In the final version of the paper, we will add a brief discussion on these possible extensions.
> > >
> > > ***References***
> > >
> > > [R1] Kirichenko, Polina, Pavel Izmailov, and Andrew Gordon Wilson. "Why normalizing flows fail to detect out-of-distribution data." Advances in neural information processing systems 33 (2020).
> > >
> > > [R2] Grover, Aditya, Manik Dhar, and Stefano Ermon. "Flow-gan: Bridging implicit and prescribed learning in generative models." arXiv preprint arXiv:1705.08868 1 (2017).

---

> ### Comment · AnonReviewer1 · 2020-11-20
> **Thanks for some of your responses**
>
> First, please respect the reviewer's time. It is not appropriate to be overly verbose and post three pages of response.
>
> 1. It is not appropriate for this paper to motivate the algorithm using criticism of memorization by GANs. The paper does not provide any experiments supporting this claim. FFs do not have any strong privacy guarantee.
> 2. You argue that the choice of $N > T$ is for theoretical lossless recovery. FFs are never guaranteed to be lossless, even when $N > T$.
> 3. Can you provide the training run times for FFs in comparison to the other methods? This result will help settle the dispute about the speed of the algorithm.
> 4. I wish you had implemented at least one of the three methods for dealing with variable length time series.
> 5. Notice that using DWT, you need to deal with spectral and temporal domains together. While it seems that FFs can be generalized to use DWT, it is not an obvious generalization.

---

### Official Review · AnonReviewer2 · 2020-10-27
**Simple but powerful new convolutional flow architecture with impressive performance on time series problems**

**Rating:** 7
**Confidence:** 4

**Review:**

Summary:
The paper introduces a new convolutional flow architecture that uses the DFT to convert the generated time series to the frequency domain. Convolutions are performed by multiplication in the frequency domain through a spectral affine layer that transform the even or odd part of the signal using a data dependent filter. The resulting time-domain convolution has input dependent weights, an interesting and original approach clearly different from other convolutional flow such as [1].

Relevance:
Time series generative modeling has a wide range of crucial applications in fields ranging from medicine to finance. The new method shows very promising performance and it has the potential to become a state-of-the-art method for time series generation.

Originality:
The use of the DFT and spectral affine layers is original in the context of normalizing flows. Importantly, the DFT is very suitable for flows since it is an isometry and has therefore a trivial Jacobin. The use of input dependent convolutions is very interesting even in the context of regular ConvNet architectures.

Scientific quality:
The method is clearly presented and well motivated. The experiment section cover a decently large set of experiments and performance are compared with a very large number of state-of-the-art baselines.

Pros:
- Original new architecture well motivated for time series applications
- Rigorous experiments with multiple relevant baselines
- Very promising experimental results

Cons:
- It would have been useful to see a wider range of real world applications

Minor points:
- I would like to have a figure with the time domain sampled generated by the FF and the other baselines. It is strange to see a generative modeling paper without a figure showing the generated samples.

References:
[1] Karami, Mahdi, et al. "Invertible convolutional flow." Advances in Neural Information Processing Systems. 2019.

---

> ### Author Response · Authors · 2020-11-17
> **Response to comments by Reviewer #2**
>
> Thank you very much for your valuable comments. Below is a description of the changes that we will apply to the final version of the paper in order to incorporate all your suggestions.
>
> ***Additional experimental results***
>
> We have added two data sets to the experimental setup in Section 5.2: (a) MIMIC-III, which is data set for intensive care unit (ICU) time-series data [R1], and (b) an electrocardiogram (ECG) data set from Kaggle [R2]. These two data sets cover two medical applications with different classes of time series data: ICU data comprises trends of biomarkers for severly ill patients over time, and ECG data comprises quasi-periodic heart's rhythm data. We hope that these additional data sets cover a wider range of applications as requested. We compared Fourier Flows with RealNVP and the most competitive baseline (TimeGAN)---results are provided below. In the final paper, we will add the performance results for all of the remaining baselines as well.
>
> _______________________________________________________
>    **Predictive scores (95% confidence intervals)**
> _______________________________________________________
>  **Model**     ----------    **MIMIC**     ----------    **ECG**
> _______________________________________________________
> TimeGAN----------0.1501 $\pm$ 0.04   ----------  0.1621 $\pm$ 0.03
>
> RealNVP-----------0.1521 $\pm$ 0.05   ----------  0.1602 $\pm$ 0.03
>
> FF--------------------0.1517 $\pm$ 0.04   ----------  0.1567 $\pm$ 0.02
> _______________________________________________________
>
> As can be seen in the Table above, FF significantly outperforms the other models on the ECG data set---this is likely because of the quasi-period nature of this data, which makes Fourier transform a natural representation for samples in ECG. FF also performs competitvely on MIMIC-III.
>
> ***Visualizing synthetic data samples***
>
> Thank you for this important suggestion. Samples from the sinusodial data set (Sec 5.1), the stock data and the ECG data (Sec 5.2) can be viewed through this anonymized link: https://ibb.co/wMkM2GX . We will add these visualizations to the supplementary material of the final version of the paper.
>
> ***References***
>
> [R1] Johnson, Alistair EW, et al. "MIMIC-III, a freely accessible critical care database." Scientific data 3.1 (2016): 1-9.
>
> [R2] https://www.kaggle.com/shayanfazeli/heartbeat/tasks

---

### Official Review · AnonReviewer3 · 2020-10-28
**The Fourier Flow approach is novel and the paper is well-written.**

**Rating:** 6
**Confidence:** 3

**Review:**

Pros:
+ This paper proposed a novel generative model named Fourier flow for modeling time series data. The model incorporates Fourier transformation into normalizing flows and considers the time series on the frequency domain, rather than the time domain. Such a combination looks interesting and novel. I think this paper will have some impact on the field of normalizing flows and time series analysis.
+ The writing of this paper looks good, which makes it easy to follow.

Cons:
- The empirical study is somewhat weak. The experimental results are not very impressive in the paper. The improvement of the performance seems to be marginal. Only one real-world dataset is considered. Please give a short description of the dataset. As the proposed method is for the general time series, the authors are suggested to evaluate their method on more datasets, especially from various fields. Besides, please give a definition of the metric predictive score.
- The paper mentions that the computational cost is no larger than some SOTAmethods. It would be better to give a discussion on the complexities of your method and the SOTAs. It would also be more convincing if the authors can show the running time in the experiment.

---

> ### Author Response · Authors · 2020-11-17
> **Response to comments by Reviewer #3**
>
> Thank you very much for your insightful review! Below is a detailed response to your comments and a description of how we will incorporate all your suggestions in the final version of the paper.
>
> ***Improvements in predictive performance***
>
> Please note that the main goal of the paper was to develop a first explicit likelihood model for time series data that is trained via exact likelihood maximization. Thus, while it is true that the performance improvements (with respect to the predictive power of an RNN model trained on synthetic data) on most data sets were marginal, the key message of our paper is that it is possible to use explicit likelihood generative models that **perform competitively** with SOTA implicit likelihood models such as GANs. We believe that this is a significant result for three reasons:
>
> - To the best of our knowledge, this is the first generative model for time series data that can be optimized using the **exact** likelihood. This is an advancement on existing explicit likelihood models, such as HMMs (Beal et al. (2002)) or Deep state space models  (Krishnan et al. (2017)), which can only be optimized using variational lower bounds.
>
> - Unlike SOTA implicit likelihood models, such as GANs, our model's accuracy (goodness-of-fit) can be evaluated using the exact model likelihood.
>
> - Unlike SOTA implicit likelihood models, such as GANs, explicit likelihood models such as ours are less prone to data memorization, which is suitable for applications with strict privacy requirements.
>
>
> ***Expanding the experimental results***
>
> First off, we would like to highlight that in our original submission, we **have considered three real-world data sets (Energy, Stocks and Lung cancer)** and not just one as indicated in your review.
>
> In addition to these 3 data sets, we have also added two extra data sets to the experimental setup in Section 5.2: (a) MIMIC-III, which is data set for intensive care unit (ICU) time-series data [R1], and (b) an electrocardiogram (ECG) data set from Kaggle [R2]. These extra data sets cover two medical applications with different classes of time series data: ICU data comprises trends of biomarkers for severly ill patients over time, and ECG data comprises quasi-periodic heart's rhythm data. We hope that these extra data sets address your request for a strengthened empirical study and widened application fields. Performance comparison with the most competitive baseline (TimeGAN) is provided below. In the final paper, we will add the performance results for all of the remaining baselines as well.
>
> _______________________________________________________
>    **Predictive scores (95% confidence intervals)**
> _______________________________________________________
>  **Model**     ----------    **MIMIC**     ----------    **ECG**
> _______________________________________________________
> TimeGAN----------0.1501 $\pm$ 0.04   ----------  0.1621 $\pm$ 0.03
>
> RealNVP-----------0.1521 $\pm$ 0.05   ----------  0.1602 $\pm$ 0.03
>
> FF--------------------0.1517 $\pm$ 0.04   ----------  0.1567 $\pm$ 0.02
> _______________________________________________________
>
>
> ***Description of the data set and evaluation metrics***
>
> Please note that, as mentioned in the paper, we have replicated the experimental setup in (Yoon et al. (2019))---the detailed description of the data sets and evaluation metrics were provided in this paper. However, to address this comment, we will add the following descriptions to the supplementary material in the final version of our paper.
>
> _______________________________________________________
> Dataset   ----------- Number of Sequences   ----------- Dimensions   ----------- Avg. Sequence Length
> _______________________________________________________
> Stocks   --------------------  3,773   --------------------  6   --------------------  24
>
> Energy   ------------------- 19,711   --------------------  29   --------------------  24
>
> Lung cancer   ----------- 149,967   --------------------  54   --------------------  58
> _______________________________________________________
>
> The evaluation metric was the defined as the mean absolute error (MAE) of the predictions made by a model trained on the synthetic data and tested on the real data. The predictive model for all baseline was 2-layer LSTM model for sequence prediction, trained to predict the next observation at each time step as highlighted in (Yoon et al. (2019)).

---

> > ### Author Response · Authors · 2020-11-17
> > **Response to comments by Reviewer #3 (cont'd)**
> >
> > ***Computational complexity***
> >
> > Thank you for this suggestion. Please recall that one of the main motivations behind using the Fourier transform in our flow is the computational efficiency of the FFT algorithm. As discussed in the paper, the Jacobian computation for the DFT is $O(1)$, whereas the computation of the DFT itself is linearithmetic, e.g. $O(N log N)$. This makes the computationally complexity of the overall procedure in the same order of the complexity for normalizing flows SOTA.
> >
> > To demonstrate the runtime complexity of our model, we compared the runtime of FF, RealNVP (the normalizing flow SOTA) and TimeGAN on the synthetic sinusoisal data set in the Table below. As we can see, FF and RealNVP run in roughly the same amount of time, whereas both methods train much more faster than the TimeGAN baseline. In the final version of the paper, we will add runtime information for all data sets under consideration.
> >
> > _____________________________________________________
> >        Total training runtime (sinusoid data set)
> > _____________________________________________________
> > TimeGAN -------------------------- 5611 sec
> >
> > RealNVP   -------------------------- 1123 sec
> >
> > FF              -------------------------- 1263 sec
> > _____________________________________________________
> >
> >
> > ***References***
> >
> > [R1] Johnson, Alistair EW, et al. "MIMIC-III, a freely accessible critical care database." Scientific data 3.1 (2016): 1-9.
> >
> > [R2] https://www.kaggle.com/shayanfazeli/heartbeat/tasks

---

### Official Review · AnonReviewer4 · 2020-10-28
**Interesting flow-based generative model for time-series**

**Rating:** 7
**Confidence:** 4

**Review:**

The authors propose a flow generative model for time-series in the Fourier domain. The time-series data are first converted to the Fourier domain. Instead of the affine coupling layer previously presented in literature, the authors have designed a frequency domain version of the same.

Pros:
1. The paper is well-written and is largely easy to follow.
2. Operating in the Fourier domain as well the affine coupling layer proposed in the paper are interesting novel contributions in the context of designing flow models for time-series.
3. Based on the experiments presented, Fourier Flow is surprisingly effective when compared to RealNVP, and also slightly outperforms other methods like TimeGAN.

Cons:
1. The authors should try to explain what it is about operating in the Fourier domain causes the improvements in prediction error. As the Fourier transform is invertible, I would imagine that the novel affine coupling layer has a larger role to play in this. This is also what the authors seem to suggest in Section 4.3. It would be great if the authors could add a baseline experiment operating in the time domain with the proposed affine layer (Equation (8)).
2. In terms of the explanations in the paper, a lot of space has been devoted to explain the Fourier transform, its properties and how to compute it efficiently. While it is done well, I feel these are very well known facts. For the determinant of the Fourier matrix, we can just use the fact that it is unitary and we immediately have |det(W)| = 1. Instead, additional experiments analysing the experiments could be more useful, such as understanding the role of network depth etc. for this particular method, or additional t-SNE plots for the remaining datasets.
3. As in all papers such as this one, it is not easy to see how the baseline approaches have been trained and how well the hyperparameters have been tuned.

Based on the author response, I am willing to increase my rating for this paper.

UPDATE AFTER AUTHOR RESPONSE:

The authors have addressed all my queries and made the changes that I requested. I am increasing my rating reflecting this. The paper has novelty, good experiments and improved performance. I would like to see the paper accepted.

---

> ### Author Response · Authors · 2020-11-17
> **Response to comments by Reviewer #4**
>
> Thank you very much for your insightful review!
>
> In what follows, we address all your comments, and highlight the changes that will be applied to the final version of the paper in order to incorporate your suggestions.
>
> ***Explaining the source of improvements in predictive accuracy***
>
> As you pointed out, and as evident in both Figure 2(b) and Table 2, Fourier Flows improved the predictive accuracy of models trained on sinusoidal synthetic data; this improvement is more notable in the synthetic data experiment (Figure 2(b)) than in the real data experiments (Table 2).
>
> Please note that the key idea behind the Fourier flow is to use a unitary transformation (i.e. one with a low-complexity/trivial Jacobean) that can then be followed by other transformations that are not bound to the strict structural assumptions adopted in the literature. The combination of DFT + a subsequent transformation (even as simple as an affine transform) creates an overall complex transformation. **We are highlighting this point to stress that Fourier transofrm is not a redundant step, and that the spectral filtering layer by itself would reduce to a standard RealNVP model (with an RNN instead of an NN) without the DFT layer.**
>
> As for the improvement in predictive accuracy, we believe that this comes from three sources, which we explain as follows:
>
> 1) *DFT layer compresses temporal information into a low-dimensional spectral representation*
>
> The first modeling advantage in a Fourier Flow is that the DFT layer compresses temporal patterns into lower-dimensional spectral patterns, enabling a more efficient distribution learning. Consider for instance the sinusoidal example in Sec 5.1. In this example the data $x$ is drawn from the following stochastic process:
>
> $x \sim \sin(2\pi\,f\,t),$ where $f$ is a random frequency drawn from a predefined distribution.
>
> To model length-$T$ time series drawn from such process using conventional methods, we would need to model a $T$-dimensional random variable. However, using the DFT transform, we would be modeling the spectral representation of $x$ given by:
>
> $X = \frac{j}{2} [\delta(v + f) - \delta(v - f)],$
>
> where $\delta$ is the Dirac-delta function. Thus, the spectral representation $X$ can be fully described with one piece of information, which is the location of the frequency component $f$. This means that modeling the distribution of $X$ simply reduces to modeling a 1-dimensional random variable, which is much more efficient than modeling a $T$-dimensional variable. This is the reason why our model significantly outeprforms RealNVP in the task of modeling random sine waves in Section 5.1.
>
> While the sinusoidal example (or periodic signals in general) is an extreme case wherein a single variable describes the entire time series, a spectral representation of a time series will in general be a low-dimensional compression of repetitive temporal patterns and trends.
>
> 2) *Fourier flows enable constructing complex transformations at no extra cost for Jacobean computation*
>
> As explained above, even without the novel spectral layer, a DFT layer followed by a simple transform (such as an affine transform) would amount for a complex overall transform without any extra complications associated with the Jacobean transformation. Also, because the DFT layer parameters are not learnable, the Fourier flow does not involve more parameters than an equivalent time-domain flow that uses the same filtering layer $h$.
>
> 3) *The novel spectral filtering layer*
>
> As you rightly point out, the novel spectral filtering layer used after the DFT step is another source of modeling improvement.
>
> We will highlight the three points above in the final version of the paper.

---

> > ### Author Response · Authors · 2020-11-17
> > **Response to comments by Reviewer #4 (cont'd)**
> >
> > ***Additional experiments***
> >
> > While it is true that many of the readers may be already familiar with the Fourier transform, many others may not be as familiar, and we feel that it is essential to provide sufficient introductory material to the topic in order for readers to fully understand the inner workings of our model.
> >
> > However, we will address your comment on the need for additional experiments and will add the following extra results both in Section 5 and in the supplementary material of the final version of the paper.
> >
> > 1)  *Additional real-world data sets*
> >
> > In the final version of the paper, we will add two extra data sets to the experimental setup in Section 5.2: (a) MIMIC-III, which is data set for intensive care unit (ICU) time-series data, and (b) an electrocardiogram (ECG) data set public available on Kaggle. Please refer for our responses to Reviewers 2 and 3 for more details regarding these extra experiments. Moreover, in the final version of the paper, we will add t-SNE plots for the remaining datasets.
> >
> > 2) *Investigating the effect of network depth on performance*
> >
> > We have re-run the experiments in Sec. 5.1 to assess the performance of our model at different number of hidden units for the spectral filtering layer. The results indicate that deeper networks only improve slightly on the accuracy of the learned time series distribution. These results will be added to the final supplementary material.
> >
> > 3) *Investigating the performance of a time-domain benchmark with novel spectral filtering layer*
> >
> > Please note that the RealNVP model used in our experiments already utilizes the same BiRNN for its transformation layer, with the exception of having BiRNN applied over time instead of its application on the spectral components as in Fourier flows. In fact, the RealNVP baseline in our experiments is exactly equivalent to a FF but without the DFT layer. This was already mentioned in the first paragraph of Section 5.1 of the original submission, but we will further highlight it in the final version of the paper.
> >
> >
> > ***Baselines' training and hyperparameter tuning***
> >
> > Regarding hyperparameter tuning, we used the hyperparameters settings for all baselines as in the supplementary material of (Yoon et al. (2019)), which was optimized for the same datasets used in our experiments. We will provide the values of all hyperparameters in the supplementary material.

---

> ### Comment · AnonReviewer4 · 2020-11-20
> **Thank you for the response**
>
> The authors have responded to all my queries adequately.
>
> Based on their response as to the source of performance improvements, as well as comments by the other reviewers, a good extension would be to consider the wavelet transform in the place of the Fourier transform. Or even a combination of the usual temporal representation (as in RealNVP) with additional branches with Fourier and wavelet transforms before the affine coupling layers.

---

### Comment · Area_Chair1 · 2020-11-04
**A few questions to the authors**

I have a few questions in addition to the ones the reviewers raised.

- Figure 1 shows that the imaginary part is unchanged by the spectral filtering.  Does this mean that the odd part of the generated time series has fourier coefficients drawn from the prior without affected by the flow?  If this would be true, I don't understand why FF works for datasets that have non-trivial odd parts.

- I suppose in Table1, the last equation in the spectral filtering/inverse function cell should be $\hat{X} = Re(\hat{X}) + j Y_2$ because $Im(\hat{X})$ is not defined.  Is the last equation in Eq.(8) correct?  Since y1 is the even part and y2 is the odd part, it should be y = y1+y2.  Actually I doubt that I misunderstand some of these parts, and came to the conclusion that the odd part is unchanged in the question above.  Perhaps what is unclear is how Y = concatenate(Y1, Y2) is passed to and treated in the next spectral filtering layer.  I understood that in the next layer, the first part of Y, i.e. Y1, is again treated as $Re(\hat{X})$ and Y2 as $Im(\hat{X})$.  Maybe I'm wrong, and the real and the imaginary parts are somehow mixed over multiple layers.  Please make it clear how the original latent variable Y drawn from the prior is transformed to the time-series in the output through multiple layers.

- In Table 2, RealNVP and FF give much better results than the original on the "Energy" dataset.  Are those results right?  Can you explain why it can happen?  Is the evaluation appropriate?

Minor suggestions (not need to reply in the rebuttal)
- FF also has also no privacy guarantee.  The observation that FF learns broad distribution doesn't mean it doesn't memorize particular data.  If the authors cannot show stronger evidence on the privacy issue, the statement should be weakened.
- In p.2 "we model the distribution p(T) independently following ..."  Independently from what?  In Eq.(2), x and T are not necessarily (and shouldn't be) independent.
- The data generation process should be explicitly written.  What prior for Y or ($Y^{(0)}$ for multilayer case) is used?
- Related to one of the questions the reviewers raised.  Can you somehow use FF for missing value imputation?  It would be nice to show superresolution performance by using the current method.  Also, I'd like to hear your thoughts on how to make FF cope with irregular missing values, which should often happen in medical data.

---

> ### Author Response · Authors · 2020-11-24
> **Thank you for your comments**
>
> Thanks a lot for your comments, it is really appreciated. We apologize for not incorporating your suggestions into the revised version as this comment was made visible later in time.
>
> Below are our responses to your comments:
>
> - It is true that our flow passes the imaginary part of the time series as is to $Y$. However, the imaginary part is already encoded into the spectral filter itself, hence both the real and imaginary parts are affected by and represented in the transformation. (i.e., the concatenated vector $Y$ does not comprise "real" and "imaginary" parts, but rather a $2D$-dimensional real-valued vector that constitutes the new space. The first component in $Y$, $Y_1$, depends on both real and imaginary parts). The reason we keep the imaginary part unchanged is to ensure that the spectral layer's Jacobean is easy to compute (i.e. a diagonal matrix). This approach is equivalent to the time domain approach used in GLOW, which passes have the feature dimensions as is and filters the other half only.
>
> Also please note that when cascading many flows, we alternate between applying the spectral filter to the real and imaginary parts. Since we never practically use 1 flow for the transformation, both real and imaginary components get used for spectral filtering every other cascaded flow.
>
> - You are right, $\hat{X} = Re(\hat{X}) + j Y_2$ because $Y_2 = Im(\hat{X})$. The confusion happens because we should have made it clearer in the table that $Im(\hat{X})$ is set to the $Y_2$ subset of the input $Y$ to the inverse function.
>
> Eq. (8) is correct because, as explained in the previous point, $Y$ is $2D$-dimensional real valued vector that concatenates the values of the $D$-dimensional  real and imaginary components output from the spectral filtering layer. For instance, if the DFT is 1 scalar complex value 2+3j, then $Y$ is a 2-D vector [2*H(3), 3].
>
> We hope that this clarifies the transformation function and resolves the confusion.
>
> - Yes, we have also noticed that the result on the Energy data set is much better than other data sets. Also, it seems that both RealNVP and FF perform well on the data set, which may suggest a discrepancy between the evaluation of flow models and other baselines. We have used the exact same legacy code in (Yoon et al 2019) to evaluate our models, but since the evaluation metric itself involves training an LSTM model, it might be that the evaluation models trained in the runs involved in evaluating our model are drastically better. We will investigate this issue further and make sure the performance gap is explained, and will exclude this dataset should we remain unsure about the explanation.
>
> Thanks a lot for your suggestions; all will be considered in the final version of the paper.
>
> - We will tone down this statement. (The same concern is echoed by reviewer 1.)
> - "Independent" may not be the best word to describe P(X, T), this statement will be replaced to reflect that T is considered as the parent node in the graphical model.
> - We use a normal distribution as the "prior", which is standard in almost all flow models. We will clarify this.
> - Both missing data and irregular sampling (which can be thought of as equivalent in time series) can be handled simply by assuming that the time domain signal is multiplied by a train of delta functions $\sum_{k \in K} \delta(t - k)$, where $K$ is the set of time steps with non-missing data. We will add a note on this and show how missing data can be conceptualized as another form of spectral filtering, where the filter is the DFT of $\sum_{k \in K} \delta(t - k)$ (which also happens to be another train of delta functions in the frequency domain).
>
> Thanks again for your feedback.

---

### Decision · Program_Chairs · 2021-01-07
**Final Decision**

**Decision:**

Accept (Poster)

**Comment:**

Nice ideas with practical advantages.